# Modeling SARS-CoV-2 antibody seroprevalence and its determinants in Ghana: A nationally representative cross-sectional survey

Irene Owusu Donkor[1]*, Sedzro Kojo Mensah[1], Duah Dwomoh[2], Jewelna Akorli[3], Benjamin Abuaku[1], Yvonne Ashong[3], Millicent Opoku[3], Nana Efua Andoh[3], Jeffrey Gabriel Sumboh[3], Sally-Ann Ohene[4], Ama Akyampomaa Owusu-Asare[4], Joseph Quartey[3], Edward Dumashie[3], Elvis Suatey Lomotey[3], Daniel Adjei Odumang[3], Grace Opoku Gyamfi[1], Christopher Dorcoo[1], Millicent Selassie Afatodzie[3], Dickson Osabutey[1], Rahmat bint Yussif Ismail[3], Isaac Quaye[3], Samuel Bosomprah[2], Vincent Munster[5], Kwadwo Ansah Koram[1]

1 Epidemiology Department, Noguchi Memorial Institute for Medical Research, University of Ghana, Legon, Ghana, 2 Department of Biostatistics, School of Public Health, University of Ghana, Legon, Ghana, 3 Parasitology Department, Noguchi Memorial Institute for Medical Research, University of Ghana, Legon, Ghana, 4 Emergency Preparedness and Response Unit, World Health Organization, Country Office, Accra, Ghana, 5 Virus Ecology Section, Laboratory of Virology, Rocky Mountain Laboratories, National Institute of Allergy and Infectious Diseases, National Institutes of Health, Hamilton, Montana, United States of America

* iowusu@noguchi.ug.edu.gh

**Data Availability Statement:** All data generated or analysed during this study are included in this published article.

## Abstract

Estimates of SARS-CoV-2 transmission rates have significant public health policy implications since they shed light on the severity of illness in various groups and aid in strategic deployment of diagnostics, treatment and vaccination. Population-based investigations have not been conducted in Ghana to identify the seroprevalence of SARS-CoV-2. We conducted an age stratified nationally representative household study to determine the seroprevalence of SARS-CoV-2 and identify risk factors between February and December 2021. Study participants, 5 years and older regardless of prior or current infection COVID-19 infection from across Ghana were included in the study. Data on sociodemographic characteristics, contact with an individual with COVID-19-related symptoms, history of COVID-19-related illness, and adherence to infection prevention measures were collected. Serum obtained was tested for total antibodies with the WANTAI ELISA kit. The presence of antibodies against SAR-COV-2 was detected in 3,476 of 5,348 participants, indicating a seroprevalence of 67.10% (95% CI: 63.71–66.26). Males had lower seroprevalence (65.8% [95% CI: 63.5–68.04]) than females (68.4% [95% CI: 66.10–69.92]). Seroprevalence was lowest in >20 years (64.8% [95% CI: 62.36–67.19]) and highest among young adults; 20–39 years (71.1% [95% CI 68.83,73.39]). Seropositivity was associated with education, employment status and geographic location. Vaccination status in the study population was 10%. Exposure is more likely in urban than rural areas thus infection prevention protocols must be encouraged and maintained. Also, promoting vaccination in target groups and in rural areas is necessary to curb transmission of the virus.

**Funding:** The study was funded by The Bill and Melinda Gates Foundation, (Investment ID INV-024130), the World Health Organization (Reference 2021/1166179-0) and the African Academy of Sciences (SARSCov2-4-20-004) to IOD. The funders had no role in study design, data collection and analysis and decision to publish or preparation of the manuscript.

**Competing interests:** The authors have declared that no competing interests exist.

## Introduction

COVID-19, caused by SARS-CoV-2, has established a global pandemic. Globally, as of 24[th] January, 2023, there had been 753,479,439 confirmed cases of COVID-19, including 6,812,798 deaths, reported to the World Health Organization (WHO), representing a case-fatality ratio of 0.90% [1]. As of 1[st] February 2023, a total of 13.26 billion vaccine doses have been administered giving a vaccine coverage of 69.4% globally. 1.18 million doses of vaccines are administered each day with 26.4% of people in low and middle income countries having received at least one dose. In Africa, there are now 9,478,533 COVID-19 reported cases, with 175,247 deaths as of 24[th] January 2023. In Ghana, the Ghana Health Service reported 171,112 COVID-19 cases and 1,462 deaths as of 30[th] January 2023 and 23,226,767 vaccine doses have been administered. Although the pandemic initially seemed to have stabilized due to naturally acquired population immunity and vaccine rollout, the disease has been characterized by new waves of infection and the development of more transmissible variants, such as the Delta and Omicron variants.

The true extent of SARS-CoV-2 infections in Ghana is likely to be greater than reported. Many people with SARS-CoV-2 infection do not come to the attention of the health system because a large proportion have asymptomatic infections and most symptomatic people have only a mild clinical illness [2, 3] COVID-19 symptoms overlap with those of other common upper respiratory tract infections that are usually self-limiting [4]. Furthermore, limited testing capacity and surveillance system gaps are likely to have contributed to under-ascertainment of SARS-CoV-2 infections in Ghana. This situation is like other parts of the world; serological studies from the USA, Spain, and Brazil identified an order of magnitude or more difference between laboratory-confirmed case counts and community infections [5–8].

Estimates of the SARS-CoV-2 transmission rate have important implications for public health policy as they provide insights into population-specific disease severity; and inform the strategic deployment of testing, therapies, and vaccines [9]. However, only a few SARS-CoV-2 antibody seroprevalence studies have been carried out in sub-Saharan Africa to date. For the few seroprevalence studies carried out in the sub-region, the focus has been on a subsection of the population [10, 11]. To improve the outcome of COVID-19 infection and reduce transmission, there is the need for detection of asymptomatic and mildly infected individuals within the population. This will help determine the number of true infections within the general population. It also provides the denominator for estimation of reproductive number, an indicator needed to assess the effectiveness of various COVID-19-related interventions [12, 13].

Currently, serological methods to diagnose infection have been developed and validated. These assays have helped provide valuable information in understanding the extent of transmission of COVID-19 and monitor how these changes occur over time. Seroprevalence surveys are needed globally and especially within Africa to understand the extent of COVID-19 spread, its impact on the public health system and build effective surveillance systems. This study aimed to estimate prevalence of SARS-CoV-2 and associated risk factors in Ghana using a nationally representative household sample.

## Methods

### Study design and participants

We conducted a multistage, cross-sectional cluster-sample survey of households in Ghana between February to December 2021. Sample collection was however done at three timepoints, February–March, May-July and November-December. Cities, towns and villages were selected in each of the transmission zones (i.e., High, Medium, Low and Very low burden) across Ghana (Fig 1). These were selected on the basis of high number of cases reported in the cities

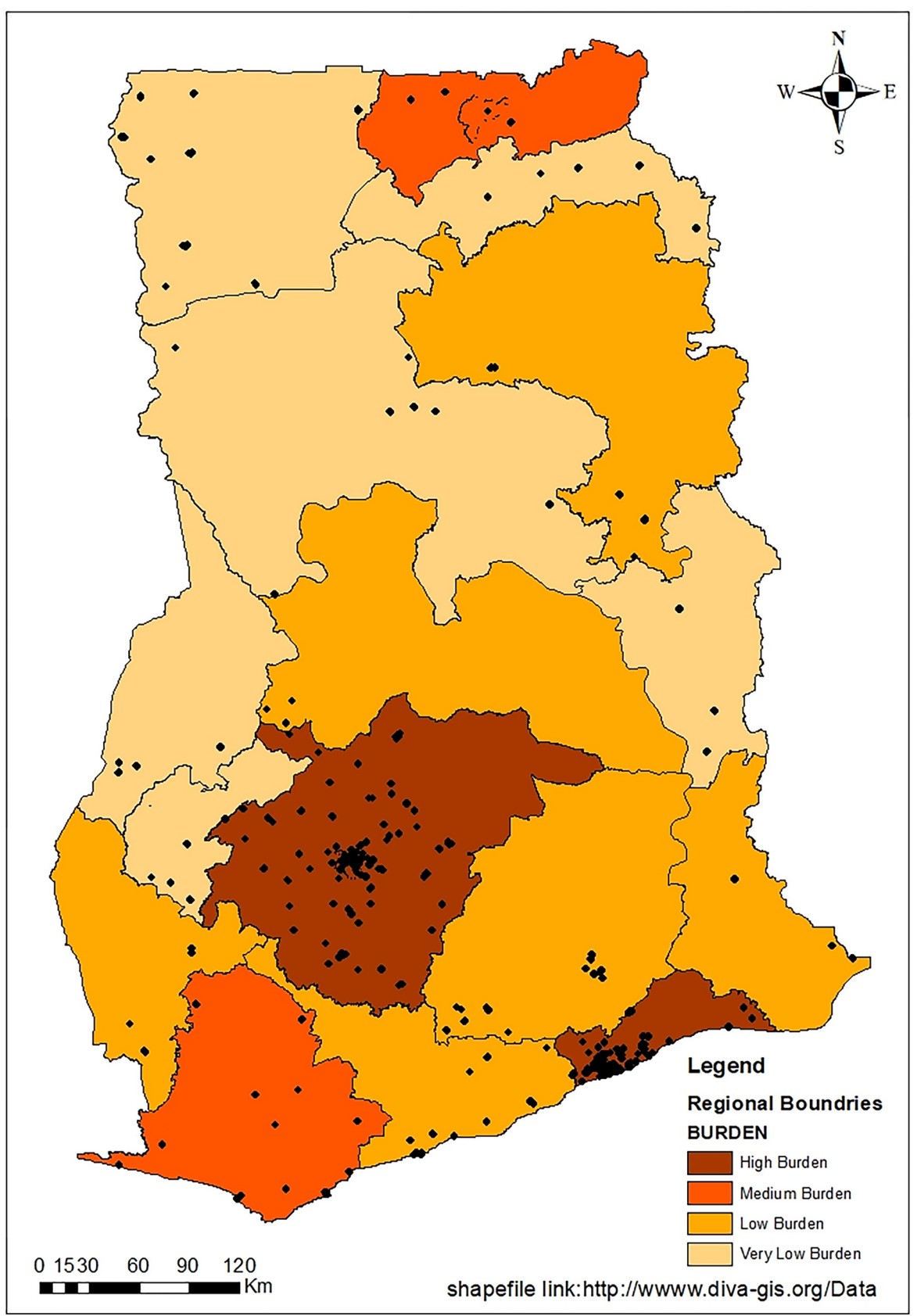

**Fig 1. Map showing areas included in the COVID-19 survey across Ghana.** The Greater Accra and Ashanti regions were the most densely populated regions and had the highest number of COVID-19 cases.

and because they are densely populated areas. Villages which are mainly rural were included to ensure that cases which may go unreported are captured and points of transmission within these areas are captured. The inclusion of villages was critical to the study as they provided information on the seroprevalence in rural Ghana.

The Ghana Statistical Service (GSS) 2021 Population and Housing Census (PHC) constituted the sampling frame. An Enumeration Area (EA) is a geographic area that covers an average of 180 households. The sampling frame contains information about the EA's location, type of residence (urban or rural), and estimated number of residential populations. In each transmission zone, cities including the peri-urbans areas, towns and villages were selected for the survey. In the first stage,122 EAs (41 EAs in Coastal, 41 in Middle and 40 in Northern) were selected with probability proportional to the percentage distribution of cities, towns and villages in the respective zones. In the second stage, an estimated number of 6000 households were visited, where 20 households per a cluster were visited. Individuals within the household aged 5 years and older in the age groups 5–9; 10–19, 20–39, 40–59 and 60+ were listed and the kish grid technique of selection was used to choose individuals for interview. One person was sampled in each household. Individuals were excluded if they refused or were unable to give informed consent; did not accept venipuncture; and were in prisons, camps, boarding houses, hostels, and hotels. All participants gave written informed consent before they were included in the study. The study design was aligned to the WHO Unity Studies Protocol.

## Sample size

The sample size was calculated to estimate SARS-CoV-2 prevalence with certain level of precision. We assumed a seroprevalence of 5% in the general population, a margin of error of 5%, a non-response rate of 10% and a design effect of 4. The country was stratified into four transmission zones and by age-class distribution (5–9; 10–19, 20–39, 40–59 and 60+) resulting into 5 strata, we arrived at a sample size 6000 households using the following formula:

$$n = Deff \times strata \times \frac{Z^2_{1-\frac{\alpha}{2}}P(1-P)}{e^2(1-NR)}$$

Where *Deff* is the designed effect, *strata* is the number of age-ecological zone-specific strata, $Z^2_{1-\frac{\alpha}{2}}$ is the square of the standard normal variate, *NR* is the non-response rate, *P* is the seroprevalence of SARS-CoV-2.

## Survey procedure

The study utilized Computer-Assisted Personal Interviewing (CAPI). The data collection application was built on Kobo Collect Tool, with data stored on Noguchi Memorial Institute for Medical Research (NMIMR) server. Six teams comprising two interviewers, two phlebotomists, and one supervisor each collected the data. The fieldwork commenced in February 2021 and was completed in December 2021. Following informed consent, participants who met the inclusion criteria were asked about their demographics, symptoms (e.g., fever, cough, shortness of breath, sore throat), exposure history, vaccination status, knowledge and attitude regarding Covid-19. The performance of field surveys was monitored using distribution checks, duration of survey, and GPS location audits. The quality of data was examined daily.

## Laboratory procedure

Samples of nasal and oropharyngeal swabs, as well as 5 mL of venous blood, were taken from each participant. Swabs were transferred in viral transport medium (VTM) to the laboratory while venous blood were separated in into whole blood, serum and plasma. Samples collected were transferred on ice to the NMIMR laboratory for processing and analyses.

Viral RNA was extracted from collected swabs in 96-well plates with the QIAamp Viral RNA Mini Kit (Qiagen Str, Hilden, Germany) using a vacuum pump extractor. The detection of the virus in a sample was a two-step process involving a screening assay targeting the SARS-CoV-2 E and N genes and confirmatory assay targeting the RNA-dependent RNA polymerase gene (RdRp), a gene specific to SARS-CoV-2. Quality controls provided in the TIB MOLBIOL kit was included in all assay plates. Controls included a positive control for E gene, RdRp gene and a negative control to detect contamination. The validity of the test was only accepted if the cycle threshold (Ct) value of the E+N and RdRp positive controls was < 36, that for the EAV was < 33 and if the NTC did not generate an amplification curve. A sample was considered conclusively positive in valid tests if the Ct values of E+N and RdRp are < 36. If the RdRp gene was not observed or the Ct > 40, the sample was considered probable for SARS. A sample was considered negative if no amplification curve was observed for the E+N and RdRp genes.

The WHO WANTAI SARS-CoV-2 total antibody kit, a sandwich Enzyme Linked Immunosorbent Assay (ELISA), was used for the qualitative detection of total antibodies to SARS-CoV-2 in each processed serum specimen. This kit has a reported sensitivity of 94.4% and a specificity of 100% with a PPV of 100% and an NPV of 95.3%. The assay is a two-step procedure using pre-coated recombinant SARS-CoV-2 antigen to capture specific SARS-CoV-2 antibodies in the serum if present. A second recombinant SARS-CoV-2 antigen conjugated to the enzyme Horseradish Peroxidase (HRP-Conjugate) is added to the immunocomplex. The absorbance of detectable antibody captured in the complete immunocomplex was measured with a BioTek Microplate Reader (Gen 5 3.10). The results were calculated by relating each specimen absorbance value to the cut-off value of the test plate. The cut-off reading was based on single filter plate reader thus, the results were calculated by subtracting the blank well absorbance value from the absorbance values of specimens and controls. Specimen giving absorbance value less than the cut-off value were considered negative. Specimen giving absorbance equal or greater to the cut-off value were considered reactive, which implies serological presence of SARS-CoV-2 antibodies. The test was accepted as positive if the absorbance value was > 1 and negative if < 1.

## Statistical analysis

The finalized data were exported to Stata 17 MP4 (StataCorp, College Station, TX, USA) for cleaning, coding, and further statistical analysis.

Weighted and unweighted frequency and percentages were utilized to describe categorical variables. The Shapiro-Wilks test, the skewness test, and the kurtosis test were performed to assess the normality of continuous variables. The median and interquartile range were used to describe and explain non-normally distributed variables. Missing values of variables were examined and confirmed to be valid. A composite adherence score was computed based on nine self-reported observations of WHO covid protocol parameters. To distinguish between low, moderate, and high levels of adherence, we developed a four-level response category level variable with scores of 0 indicating no adherence, 1–2 indicating low adherence, 3–6 indicating moderate adherence, and 7 to 9 indicating high adherence.

The seroprevalence of SARS-CoV-2 infection was calculated using 95% confidence intervals for unvaccinated individuals. The number of individual positive test results were divided by the total number of tests done overall and per stratum, during the survey period using the appropriate sampling weights and considering the sampling strategy used for the survey.

The stratification of seroprevalence was also used to describe the difference between total antibody seropositivity and demographic characteristics, adherence to Covid-19 preventative measures, and symptoms. The Pearson's Chi-square tests was utilized to determine the differences between seroprevalence statuses. Variables associated with the total antibody seropositivity, that had a likelihood ratio p-value < 0.5 were included in fitting the multivariate models.

Multiple binary logistic and Poisson regression models with a robust estimate of the variance were used to identify the factors associated with total antibody positivity using a manual stepwise selection process, and the model with the lowest Akaike information criterion (AIC) was chosen as the final parsimonious model. Adjusting for any likely confounders, odds ratios (OR) or prevalence ratios (PRs) with 95% confidence intervals (Cls) were calculated. In addition, survey design factors (stratification, clustering, and sampling weights) were accounted for in all analysis. Using variance inflation factors, multicollinearity was checked.

### Ethical consideration

Procedures in this study conform with the Ghana Public Health Act, 2012 (Act 851) and the Data Protection Act, 2012. Ethical approval (NMIMR-IRB CPN 075/19-20) was obtained from the Institutional Review Board (IRB) of the Noguchi Memorial Institute for Medical Research (NMIMR), University of Ghana. Informed consent was obtained from all subjects and/or their legal guardians (when subject is below 18 years). All consenting participants agreed to future use of their samples.

## Results

### Characteristics of the study participants by strata of districts

Table 1 shows the stratification-based distribution of sociodemographic factors, vaccination, and compliance with COVID-19 preventive measures. Of the 6000 households in 122 enumeration areas and 124 districts throughout the four strata, 5939 households consented to be surveyed. At the time of the study, nine out of ten participants who had provided informed consent were unvaccinated (n = 5,389).

### Seroprevalence of SARS-CoV-2 by sociodemographic characteristics

Of the 5,348 unvaccinated participants who participated in the survey, more than half were female (n = 3,036,56.8%, Table 2), 68.9% lived in an urban area, and 31.1% in a rural area; 49.7% of them had secondary or education or higher and 53.0% were unemployed.

The presence of total antibodies (IgG and IgM) against SAR-CoV-2 was detected in 3,476 of the 5,348 participants, indicating weighted seroprevalence of 67.10% (95% CI: 63.71–66.26) (Table 2). Males had a lower weighted seroprevalence (65.8% [95% CI: 63.5–68.04] than females (68.4% [95% CI: 66.10–69.92]) (Table 2). No significant difference ($p$ = 0.231 was found between seroprevalence and sex (Table 2). The weighted seroprevalence was lowest in those below 20 years (64.8% [95% CI: 62.36–67.19]) and highest among young adults between 20–39 years (71.1% [95% CI 68.83,73.39]). There was a significant difference between seropositivity and age groups. The weighted estimates of seropositivity and related risk factors can be seen in Table 2.

**Table 1. Background characteristics of respondents by strata of districts.**

| Characteristics | Zones (Stratum) | | | | |
|---|---|---|---|---|---|
| | Overall | High | Medium | Low | Very low |
| | N = 5,939 | N = 3,951 | N = 934 | N = 609 | N = 445 |
| **Sex** | | | | | |
| Male | 2,571 (43.3) | 1,701 (43.1) | 382 (40.9) | 274 (45.0) | 214 (48.1) |
| Female | 3,368 (56.7) | 2,250 (56.9) | 552 (59.1) | 335 (55.0) | 231 (51.9) |
| **Age(years)** | | | | | |
| Median (IQR) | 26 (15–42) | 26 (15–42) | 25 (14–42) | 26 (15–42) | 26 (15–42) |
| <20 | 2,242 (37.8) | 1,489 (37.7) | 370 (39.6) | 221 (36.3) | 162 (36.4) |
| 20–39 | 2,079 (35.0) | 1,386 (35.1) | 312 (33.4) | 220 (36.1) | 161 (36.2) |
| 40–59 | 995 (16.8) | 669 (16.9) | 153 (16.4) | 101 (16.6) | 72 (16.2) |
| 60+ | 623 (10.5) | 407 (10.3) | 99 (10.6) | 67 (11.0) | 50 (11.2) |
| **Educational level** | | | | | |
| Never attended school | 884 (14.9) | 487 (12.3) | 141 (15.1) | 102 (16.7) | 154 (34.6) |
| Primary | 1,953 (32.9) | 1,305 (33.0) | 327 (35.0) | 183 (30.0) | 138 (31.0) |
| Secondary+ | 3,102 (52.2) | 2,159 (54.6) | 466 (49.9) | 324 (53.2) | 153 (34.4) |
| **Employment status** | | | | | |
| Unemployed | 3,006 (50.6) | 2,024 (51.2) | 487 (52.1) | 300 (49.3) | 195 (43.8) |
| Employed | 2,933 (49.4) | 1,927 (48.8) | 447 (47.9) | 309 (50.7) | 250 (56.2) |
| **Vaccinated** | | | | | |
| No | 5,389 (90.7) | 3,459 (87.5) | 898 (96.1) | 592 (97.2) | 440 (98.9) |
| Yes | 550 (9.3) | 492 (12.5) | 36 (3.9) | 17 (2.8) | 5 (1.1) |
| **Symptoms** | | | | | |
| No symptoms | 4,111 (69.2) | 2,780 (70.4) | 623 (66.7) | 429 (70.4) | 279 (62.7) |
| Only 1 | 882 (14.9) | 573 (14.5) | 147 (15.7) | 88 (14.4) | 74 (16.6) |
| Two or more symptoms | 946 (15.9) | 598 (15.1) | 164 (17.6) | 92 (15.1) | 92 (20.7) |
| **Adherence to COVID-19 protocols** | | | | | |
| No adherence | 1,827 (30.8) | 937 (23.7) | 289 (30.9) | 330 (54.2) | 271 (60.9) |
| Low | 1,690 (28.5) | 1,144 (29.0) | 266 (28.5) | 204 (33.5) | 76 (17.1) |
| Moderate | 1,481 (24.9) | 1,055 (26.7) | 272 (29.1) | 67 (11.0) | 87 (19.6) |
| High | 539 (9.1) | 413 (10.5) | 107 (11.5) | 8 (1.3) | 11 (2.5) |
| Missing | 402 (6.8) | 402 (10.2) | 0 (0.0) | 0 (0.0) | 0 (0.0) |
| **Rural/Urban** | | | | | |
| Rural | 1,709 (28.8) | 862 (21.8) | 372 (39.8) | 220 (36.1) | 255 (57.3) |
| Urban | 4,230 (71.2) | 3,089 (78.2) | 562 (60.2) | 389 (63.9) | 190 (42.7) |

Data are presented as median (IQR) for continuous measures, and n (%) for categorical measures.

## Seroprevalence of SARS-CoV-2 by adherence to COVID-19 protocols

Seropositivity to SAR-COV-2 was higher among persons that did not adhere to WHO-standard prevention protocols (35.1% [95% CI: 33.5–36.8; Table 3) compared to those who adhered to the protocols (Table 3). Among those who did not show any symptoms of SARS-CoV-2 infection, 68.3% [95% CI 66.7–69.8] were seropositive to SAR-CoV-2.

**Multivariable analysis of factors associated seropositivity of SAR-COV-2.** Total antibody positivity was less prevalent in men (adjusted weighted OR: 0.93; 95 percent CI:0.81–1.07; Table 4). Working individuals were at higher risk (adjusted weighted OR:1.23, 95% CI:1.00–1.51, p<0.05), as were urban residents (adjusted weighted OR:1.46, 95% CI:1.26–1.69, p<0.001).

**Table 2. SARS-CoV-2 seroprevalence by sociodemographic characteristics and related risk factors.**

| Characteristics | Number (%) of Unvaccinated Participants | Number of Total Antibody Positive | Unweighted Seropositive % [95% CI] | Weighted Seropositive % [95% CI] | Pearson (chi2) P-value |
|---|---|---|---|---|---|
| **Overall** | **5,348** | **3,476** | **65.00[63.71,66.26]** | **67.10[65.64,68.54]** | |
| **Sex** | | | | | 0.231 |
| Male | 2,312 (43.2) | 1,482 | 64.10[63.97, 67.35]. | 65.84 [63.58,68.04]. | |
| Female | 3,036 (56.8) | 1,994 | 65.68[62.12, 66.03] | 68.04 [66.10,69.92] | |
| **Age in years** | | | | | <0.001 |
| <20 | 2,199 (41.1) | 1,379 | 62.71[60.67,64.71] | 64.81 [62.36, 67.19] | |
| 20–39 | 1,852 (34.6) | 1,278 | 69.01[66.86,71.07] | 71.16 [68.83,73.39] | |
| 40–59 | 830 (15.5) | 535 | 64.46[61.14, 67.64] | 67.00 [63.37,70.45] | |
| 60+ | 467 (8.7) | 284 | 60.81[56.31,65.14] | 70.44 [57.10,67.03] | |
| **Educational level** | | | | | <0.001 |
| Never attended school | 820 (15.3) | 476 | 58.05[54.64,61.38] | 60.43 [56.57,64.16] | |
| Primary | 1,868 (34.9) | 1,174 | 62.85[60.63, 65.01] | 65.14 [62.58,67.61] | |
| Secondary+ | 2,660 (49.7) | 1,826 | 68.65[66.86,70.38] | 68.65[68.42, 72.39] | |
| **Employment status** | | | | | 0.006 |
| Unemployed | 2,832 (53.0) | 1,793 | 63.31[61.52,65.07] | 65.16 [63.03, 67.22] | |
| Employed | 2,516 (47.0) | 1,683 | 66.89[65.03,68.70] | 69.37 [67.35, 71.32] | |
| **Symptoms** | | | | | 0.567 |
| No symptoms | 3,667 (68.6) | 2,373 | 64.71[63.15,66.24] | 66.50 [64.71, 68.24] | |
| Only 1 | 789 (14.8) | 526 | 66.67[63.30,69.87] | 68.81 [65.04, 72.35] | |
| Two or more symptoms | 892 (16.7) | 577 | 64.69[61.49,67.76] | 68.17 [64.55, 71.57] | |
| **Adherence to COVID-19 protocols** | | | | | 0.590 |
| No adherence | 1,742 (35.2) | 1,180 | 67.74[65.50,69.89] | 69.72 [67.16,72.16] | |
| Low | 1,480 (29.9) | 985 | 66.55[64.11,68.91] | 69.16 [66.46,71.73] | |
| Moderate | 1,295 (26.1) | 892 | 68.88[66.30,71.34] | 68.87 [66.03,71.57] | |
| High | 438 (8.8) | 301 | 68.72[64.22,72.89] | 69.32 [64.41,73.84] | |
| **Location** | | | | | <0.001 |
| Rural | 1,664 (31.1) | 1,028 | 61.78[59.42,64.09] | 63.04 [60.45, 65.56] | |
| Urban | 3,684 (68.9) | 2,448) | 66.45[64.91,67.96] | 68.45 [66.70, 70.16] | |
| **Zones** | | | | | 0.009 |
| High | 3,422 (64.0) | 2,235 | 65.31[61.58, 70.41] | 67.97 [66.09, 69.80] | |
| Medium | 896 (16.8) | 602 | 67.19[54.96,62.89] | 68.39 [65.03,71.57] | |
| Low | 590 (11.0) | 348 | 58.98[64.04,70.19] | 59.11 [54.82,63.26] | |
| Very low | 440 (8.2) | 291 | 66.14[63.70,66.89] | 66.67 [61.76,71.24] | |

* **Pearson** chi square test: unweighted seroprevalence.

## Discussion

Regardless of estimates based on laboratory-confirmed cases, the true population prevalence of SARS-CoV-2 total antibodies following vaccination is unknown, despite the fact that Ghana has been struck by 3 waves of the COVID-19 pandemic. This study estimates the prevalence of anti-SARS-CoV-2 antibodies in the Ghanaian population and identifies risk factors for infections by a stratified population-based survey. Serological assays were utilized to determine the serostatus of study respondents.

Results from the survey showed that 67.10 percent of Ghana's population aged 5 and older have been exposed to SARS-CoV-2 virus, estimating 18.1 million infections by December

**Table 3. Seroprevalence and adherence to WHO prevention protocols.**

| WHO prevention guidelines | Unvaccinated individuals | Number of Total Antibody Positive | Unweighted Seropositive % [95% CI] | Pearson Chi (2) p-value |
|---|---|---|---|---|
| **Washing hands more often/for longer(n = 4,955)** | | | | 0.091 |
| No | 2,461 | 1,640 | 48.84[47.15,50.53] | |
| Yes | 2,494 | 1,718 | 51.16 [49.47,52.85] | |
| **Avoiding nonessential social contact(n = 4,955)** | | | | 0.442 |
| No | 3,968 | 2,679 | 79.78 [78.39,81.10] | |
| Yes | 987 | 679 | 20.22[18.90, 21.61] | |
| **Social distancing (n = 4,955)** | | | | 0.005 |
| No | 4,070 | 2,723 | 81.09[79.73,82.38] | |
| Yes | 885 | 635 | 18.91[17.62,20.27] | |
| **Avoiding visits to public places when possible (n = 4,955)** | | | | 0.806 |
| No | 4,323 | 2,927 | 87.16[85.99, 88.25] | |
| Yes | 632 | 431 | 12.84[11.75,14.01] | |
| **Self-isolating at home(n = 4,955)** | | | | 0.996 |
| No | 4,316 | 2,925 | 87.11[85.93, 88.20] | |
| Yes | 639 | 433 | 12.89[11.80,14.07] | |
| **Using hand sanitizing gel/using it more often (n = 4,955)** | | | | 0.242 |
| No | 3,082 | 2,070 | 61.64[59.99,63.28] | |
| Yes | 1,873 | 1,288 | 38.36[36.72,40.01] | |
| **Avoiding public transport when possible (n = 4,955)** | | | | 0.367 |
| No | 4,264 | 2,900 | 86.36[85.16,87.48] | |
| Yes | 691 (13.9) | 458 | 13.64[12.52, 14.84] | |
| **Working from home/working from home more often(n = 4,955)** | | | | 0.769 |
| No | 4,332 | 2,939 | 87.52[86.36,88.60] | |
| Yes | 623 | 419 | 12.48[11.40,13.64] | |
| **Wearing a mask (n = 4,955)** | | | | 0.493 |
| No | 2,762 | 1,883 | 56.08[54.39,57.75] | |
| Yes | 2,193 | 1,475 | 43.92[42.25,45.61] | |
| **Overall adherence (n = 4,955)** | | | | 0.590 |
| No adherence | 1,742 | 1,180 | 35.14[33.54,36.77] | |
| Low | 1,480 | 985 | 29.33[27.82,30.90] | |
| Moderate | 1,295 | 892 | 26.56[25.10,28.08] | |
| High | 438 | 301 | 8.96[8.04,9.98] | |
| **Symptoms(n = 5,348)** | | | | 0.567 |
| No symptoms | 3,667 | 2,373 | 68.27[66.70,69.80] | |
| Only 1 | 789 | 526 | 15.13[13.98,16.36] | |
| Two or more symptoms | 892 | 577 | 16.60[15.40,17.87] | |

2021. Across the stratification zones, the prevalence of SARS-CoV-2 infection in the general population varied. The stratification was based on the Ghana Health Service's reported number of COVID-19 cases as of October 20,2020. Exposure in high prevalence areas is 13% higher than very low prevalence areas, however this difference is not statistically significant. Low (64%) and medium (4%) burden areas have, respectively, a reduced risk of exposure than very low burden areas.

**Table 4. Factors associated with Covid-19 seroprevalence: A multiple regression analysis.**

| | Model 1 | | Model 2 | |
|---|---|---|---|---|
| Variables | Unweighted | Weighted† | Unweighted | Weighted† |
| | aOR [95% CI] | aOR [95% CI] | aPR [95% CI] | aPR [95% CI] |
| **Sex** | | | | |
| Female | 1 | 1 | 1 | 1 |
| Male | 0.96[0.85,1.09] | 0.93[0.81,1.07] | 0.99[0.95,1.03] | 0.98[0.94,1.02] |
| **Age (Years)** | | | | |
| <20 | 1 | 1 | 1 | 1 |
| 20–39 | 1.16[0.96,1.40] | 1.17[0.94,1.46] | 1.05[0.99,1.11] | 1.05[0.98,1.12] |
| 40–59 | 1.02[0.80,1.29] | 1.02[0.78,1.35] | 1.01[0.94,1.09] | 1.01[0.93,1.10] |
| 60+ | 0.97[0.76,1.24] | 0.94[0.71,1.25] | 0.99[0.91,1.08] | 0.99[0.90,1.08] |
| **Educational level** | | | | |
| Never attended school | 1 | 1 | 1 | 1 |
| Primary | 1.22[1.01,1.46] * | 1.19[0.97,1.47] | 1.08[1.01,1.15] * | 1.06[0.99,1.15] |
| Secondary+ | 1.57[1.31,1.87] *** | 1.48[1.21,1.82] *** | 1.16[1.09,1.24] *** | 1.13[1.06,1.22] *** |
| **Employment Status** | | | | |
| Unemployed | 1 | 1 | 1 | 1 |
| Employed | 1.16[0.97,1.38] | 1.23[1.00,1.51] * | 1.05[0.99,1.10] | 1.06[1.00,1.13] * |
| **Location** | | | | |
| Rural | 1 | 1 | 1 | 1 |
| Urban | 1.47[1.29,1.68] *** | 1.46[1.26,1.69] *** | 1.14[1.09,1.19] *** | 1.13[1.08,1.18] *** |
| **Zones** | | | | |
| Very low | 1 | 1 | 1 | 1 |
| Low | 0.64[0.49,0.83] *** | 0.63[0.47,0.84] ** | 0.85[0.78,0.94] ** | 0.85[0.77,0.94] ** |
| Medium | 0.96[0.75,1.24] | 1.04[0.79,1.37] | 0.99[0.91,1.07] | 1.01[0.93,1.10] |
| High | 1.04[0.83,1.30] | 1.13[0.88,1.46] | 1.01[0.94,1.09] | 1.04[0.96,1.12] |
| **Number of symptoms** | | | | |
| No symptoms | 1 | 1 | 1 | 1 |
| Only 1 | 1.04[0.87,1.23] | 1.08[0.89,1.32] | 1.01[0.96,1.07] | 1.02[0.97,1.08] |
| Two or more symptoms | 1.06[0.90,1.26] | 1.18[0.98,1.42] | 1.02[0.97,1.08] | 1.05[0.99,1.11] |
| **Covid adherence** | | | | |
| No adherence | 1 | 1 | 1 | 1 |
| Low | 0.86[0.73,1.00] * | 0.88[0.74,1.05] | 0.95[0.91,1.00] * | 0.96[0.91,1.01] |
| Moderate | 0.89[0.76,1.05] | 0.81[0.67,0.97] * | 0.96[0.92,1.01] | 0.94[0.89,0.99] * |
| High | 0.86[0.68,1.08] | 0.80[0.61,1.04] | 0.95[0.89,1.02] | 0.94[0.87,1.01] |
| *AIC* | 6140.9 | 6010.2 | 9322.7 | 9363.5 |
| *BIC* | 6251.5 | 6120.8 | 9433.3 | 9474.1 |
| Observations | 4955 | 4955 | 4955 | 4955 |

Abbreviation: aOR: Adjusted odds ratio; aPR: Adjusted Prevalence ratio; *AIC* Akaike's information criterion; *BIC* Schwarz's information criterion; CI: Confidence interval. P-value notation

***p<0.001

**p<0.01

*p<0.05

†: Sampling weights applied.

Stratified by seroprevalence, male and female seroprevalence were comparable (65.84 percent vs. 68.04 percent), however the difference between seroprevalence and sex was not statistically significant (p = 0.231), indicating that both sexes were exposed to and vulnerable to the

virus. Similar to a study conducted in the United States of America, which found no difference between seroprevalence and sex [14], this result was observed. Weighted seroprevalence among age categories varied from 65 to 70 percent (below 20 years (64.8% [95% CI: 62.36–67.19]); between 20–39 (67.0% [95% CI: 68.83–73.39]); between 40-59(67.00% [95% CI:63.37–70.45]);60 years and beyond (70.44% [95% CI:57.10–67.03]; P<0.001).

Considering the findings of the seroprevalence, the following variables were associated with seropositivity: a person's level of education, employment status, geographic location, and stratification zone. The weighted adjusted odds of seropositivity were 23% higher among those who completed primary education versus those who did not, which is statistically significant at the 5% confidence level. Again, individuals with at least a secondary education had a 48 percent larger risk of seropositivity than those with less education. People who lived in urban areas had 46% higher seropositivity rates than those who lived in rural areas. A comparable study in India found that urban regions had a greater prevalence than rural areas [15].

Findings also indicated that adherence to COVID-19 prevention guidelines had no difference on the seroprevalence of SARS-CoV-2 infection. The risk of exposure was reduced by 88% for those who adhered as compared to non-adherents. This seroprevalence did not correlate with symptoms, indicating that a large percentage of the population is asymptomatic. There is a lower chance of infection when only one symptom is present as compared to when two or more symptoms are present (8% vs 18%). The result of adherence is in contrast to a survey in Ecuador [16], which found that adherence was related with higher rates of seroprevalence.

The study had some limitations, such as the fact that it was underpowered for regional and district analyses. Cross reactivity with other human coronaviruses and other respiratory viruses. Studies have shown high prevalence of pre-existing serological cross-reactivity against SARS-CoV-2 [17, 18] leading to high seroprevalence of the SARS-CoV-2 virus. Also excluding persons in congregate settings such as prisons and hostels may have affected population level estimates since data on transmission of the disease in these settings are not readily available due to government restrictions in such areas during the peak of the pandemic.

In conclusion, our findings indicate that seven out of ten individuals aged 5 or older were exposed to the SARS-CoV-2 virus over the time of data collection. Seroprevalence is however lower in the 60+ age group compared to the 5–59 age group. Urban areas have a higher seroprevalence than rural areas, and adherence to COVID-19 prevention protocols should be enforced to avoid the transmission of the virus. Also, seropositivity suggest prior exposure to COVID-19 or other coronaviruses which may confer some level of immunity. Therefore, those groups with lower sero-prevalence may have a higher risk of contracting COVID 19. One of the public health implications include promoting vaccination especially targeting groups with lower prevalence such as those above 60 years and rural areas.

## Acknowledgments

We acknowledge the technical support of Felix Ansah at the West African Centre for Cell Biology of Infectious Pathogens (WACCBIP) and providing us access to their Quantstudio Real-time PCR equipment. The research staff of Departments of Parasitology and Epidemiology, Mr. Sampson Otoo, Prince Horlortu, Juliana Tetteh, Prince Ato Donkor and, Essabella Scott for their involvement in field sampling. We would also like to thank Dr. Robert Fischer, Dr. Kwe Claude Yinda, Dr. Neeltje van Doremalen, and all members of staff of the Virus Ecology Section, Rocky Mountain Laboratories, Hamilton Montana for the immense support given in the conceptualization and conduct of this study. We acknowledge the support of Dr. Terrel Sanders, head of the United States Naval Medical Research Unit 3 and all his staff for the

support given in shipping and receiving reagents and consumables to conduct this study. We acknowledge the staff of the Ghana Statistical Service, Mr. Peter Peprah and his team for helping us calculate and identify enumeration areas for this study. Finally, we acknowledge the support of Nana Yaa Asiedua Appiah, Elizabeth Obeng Aboagye, Benedicta Kwablah Ayertey all of the Infectious Disease Epidemiology Laboratory (iDEL).

## Author Contributions

**Conceptualization:** Irene Owusu Donkor, Sedzro Kojo Mensah, Jewelna Akorli, Benjamin Abuaku, Vincent Munster, Kwadwo Ansah Koram.

**Data curation:** Irene Owusu Donkor, Elvis Suatey Lomotey, Daniel Adjei Odumang, Kwadwo Ansah Koram.

**Formal analysis:** Irene Owusu Donkor, Sedzro Kojo Mensah, Duah Dwomoh, Samuel Bosomprah.

**Funding acquisition:** Irene Owusu Donkor, Benjamin Abuaku, Vincent Munster, Kwadwo Ansah Koram.

**Investigation:** Irene Owusu Donkor, Sedzro Kojo Mensah, Jewelna Akorli, Benjamin Abuaku, Yvonne Ashong, Millicent Opoku, Jeffrey Gabriel Sumboh, Elvis Suatey Lomotey, Daniel Adjei Odumang, Grace Opoku Gyamfi, Christopher Dorcoo, Dickson Osabutey, Rahmat bint Yussif Ismail, Isaac Quaye, Kwadwo Ansah Koram.

**Methodology:** Irene Owusu Donkor, Sedzro Kojo Mensah, Duah Dwomoh, Jewelna Akorli, Benjamin Abuaku, Yvonne Ashong, Millicent Opoku, Nana Efua Andoh, Joseph Quartey, Edward Dumashie, Elvis Suatey Lomotey, Daniel Adjei Odumang, Grace Opoku Gyamfi, Christopher Dorcoo, Kwadwo Ansah Koram.

**Project administration:** Irene Owusu Donkor.

**Resources:** Irene Owusu Donkor.

**Software:** Sedzro Kojo Mensah.

**Supervision:** Irene Owusu Donkor, Jewelna Akorli, Yvonne Ashong, Millicent Opoku, Jeffrey Gabriel Sumboh.

**Validation:** Irene Owusu Donkor, Sedzro Kojo Mensah.

**Visualization:** Irene Owusu Donkor, Sedzro Kojo Mensah.

**Writing – original draft:** Irene Owusu Donkor, Sedzro Kojo Mensah, Duah Dwomoh, Samuel Bosomprah.

**Writing – review & editing:** Irene Owusu Donkor, Sedzro Kojo Mensah, Duah Dwomoh, Jewelna Akorli, Benjamin Abuaku, Yvonne Ashong, Millicent Opoku, Nana Efua Andoh, Jeffrey Gabriel Sumboh, Sally-Ann Ohene, Ama Akyampomaa Owusu-Asare, Joseph Quartey, Edward Dumashie, Elvis Suatey Lomotey, Daniel Adjei Odumang, Grace Opoku Gyamfi, Christopher Dorcoo, Millicent Selassie Afatodzie, Dickson Osabutey, Rahmat bint Yussif Ismail, Isaac Quaye, Samuel Bosomprah, Vincent Munster, Kwadwo Ansah Koram.

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
