## [Decision Letter · Decision Letter 0]

2 Jan 2023

PGPH-D-22-01388

Modeling SARS-CoV-2 antibody seroprevalence and its determinants in Ghana: a nationwide cross-sectional survey

Dear Dr. Donkor,

Thank you for submitting your manuscript to PLOS Global Public Health. After careful consideration, we feel that it has merit but does not fully meet PLOS Global Public Health’s publication criteria as it currently stands. Therefore, we invite you to submit a revised version of the manuscript that addresses the points raised during the review process.

We look forward to receiving your revised manuscript.

Kind regards,

Tarun Bhatnagar, MD, PhD, PGDBE

Academic Editor

Journal Requirements:

a. Please clarify all sources of funding (financial or material support) for your study. List the grants (with grant number) or organizations (with url) that supported your study, including funding received from your institution. 

b. State the initials, alongside each funding source, of each author to receive each grant.

c. State what role the funders took in the study. If the funders had no role in your study, please state: “The funders had no role in study design, data collection and analysis, decision to publish, or preparation of the manuscript.”

d. If any authors received a salary from any of your funders, please state which authors and which funders.

2. Please provide separate figure files in .tif or .eps format.

3. We do not publish any copyright or trademark symbols that usually accompany proprietary names, eg (R), (C), or TM  (e.g. next to drug or reagent names). Please remove all instances of trademark/copyright symbols throughout the text, including R on page 13.

4. Some material included in your submission may be copyrighted. According to PLOS’s copyright policy, authors who use figures or other material (e.g., graphics, clipart, maps) from another author or copyright holder must demonstrate or obtain permission to publish this material under the Creative Commons Attribution 4.0 International (CC BY 4.0) License used by PLOS journals. Please closely review the details of PLOS’s copyright requirements here: PLOS Licenses and Copyright. If you need to request permissions from a copyright holder, you may use PLOS's Copyright Content Permission form.

Potential Copyright Issues:

Figure 1: please (a) provide a direct link to the base layer of the map (i.e., the country or region border shape) and ensure this is also included in the figure legend; and (b) provide a link to the terms of use / license information for the base layer image or shapefile. We cannot publish proprietary or copyrighted maps (e.g. Google Maps, Mapquest) and the terms of use for your map base layer must be compatible with our CC-BY 4.0 license. 

Additional Editor Comments (if provided):

Reviewers' comments:

Reviewer's Responses to Questions

**Comments to the Author**

1. Does this manuscript meet PLOS Global Public Health’s publication criteria? Is the manuscript technically sound, and do the data support the conclusions? The manuscript must describe methodologically and ethically rigorous research with conclusions that are appropriately drawn based on the data presented.

Reviewer #1: Yes

Reviewer #2: Partly

2. Has the statistical analysis been performed appropriately and rigorously?

Reviewer #1: Yes

Reviewer #2: Yes

3. Have the authors made all data underlying the findings in their manuscript fully available (please refer to the Data Availability Statement at the start of the manuscript PDF file)?

Reviewer #1: Yes

Reviewer #2: No

4. Is the manuscript presented in an intelligible fashion and written in standard English?

Reviewer #1: Yes

Reviewer #2: Yes

5. Review Comments to the Author

Reviewer #1: The authors have estimated COVID-19 seroprevalence for Ghana through a cross sectional survey. I have uploaded my comments as a separate attachment. Methods section especially sampling procedures have to be mentioned clearly.

Reviewer #2: Modeling SARS-CoV-2 antibody seroprevalence and its determinants in Ghana: a nationwide cross-sectional survey

PGPH-D-22-01388

The study team has conducted a seroprevalence study and provide important insights into the potential magnitude of, and differential risks related to, SARS-CoV-2 exposure. The aim of the study is to estimate the population-level seroprevalence (as a marker for exposure) and identify risk-factors (exposure risks). The study is important and well-done, and a particular strength is the representative sampling strategy. My recommendations will hopefully serve to further improve the paper and its conclusions/implications.

Main suggestions:

The main concern with the study as currently analysed, is selection biases stemming from restricting the seroprevalence risk-factor analyses to unvaccinated (90% of the sample). It would be useful to first indicate if vaccination might influence (or not) the interpretation of the serological tests and if it is possible to differentiate serology results on the basis of vaccination. If this is not possible, then consider explicitly discussing the limitations of the risk-factor analyses given the selection biases, and explicitly show the distribution of the same covariates by vaccination status (in main text ideally, or appendix if not enough room).

It took this reviewer a little while to understand this nuance in study design: line 171 suggests seroprevalence estimates only made among persons not yet vaccinated. And thus would suggest clarifying that although sample collection was among a population-based sample, that the seroprevalence study was only among those not yet vaccinated (e.g. in the objective of the study). Similarly, eg. line 232 should specific “among unvaccinated” (subheading = Multivariable analysis of factors associated seropositivity of SAR-COV-2)

A key limitation of the risk factor analyses therefore, is that we only have risk factors for exposure among unvaccinated. This creates a selection bias when interpreting the risk factors. It would therefore be useful to show how the covariates or variables (i.e. exposure risk-factors) of interest vary by vaccination status (e.g. as an appendix table), and to describe how this limitation may effect our interpretation of the risk factors. For example, if older adults were more likely to be vaccinated, especially if they are more likely to be at higher exposure risk, then they are excluded from the seroprevalence-risk-factor analyses – leading to a potentially misleading interpretation from the seroprevalence-risk-factor model that older adults are at lower exposure risk. That is, if vaccination status is correlated with exposure risks, then the selection biases have the potential to lead to opposite (misleading) conclusions.

Given that vaccination status was self-reported, please discuss in limitations how this might affect the results of the seroprevalence analyses.

In the limitations section, it would be helpful to discuss how exclusion of specific “congregate” settings such as prisons, hostels, etc. might affect the population-level estimates, particularly if data or information available about outbreaks in such settings.

The final statistical analyses could be clarified a bit more: if the rationale for the stepwise regression was that the final model is based on the most parsimonious model (using AIC), then how are likely confounders also included? I was also confused by two models depicted in table 4.

Abstract: the conclusions would benefit from clarification. For example, the statement” … strict adherence must be maintained” is not a concluding statement that follows logically from the results. The conclusion that promoting vaccination among subsets of the population with lower seroprevalence seems counter to what external data suggest on the benefits of hybrid immunity, as well as ongoing differential risks among those with highest risks of exposures. Would recommend revising the conclusion statements in the abstract to focus on what the differential seroprevalence data suggest about differential risks, and large magnitude of exposure risks – for example, in the context of education and employment status.

Minor suggestions:

- Abstract: suggest adding information about representativeness of the sample

- Abstract: indicate the time-period of data collection

- Abstract: would be helpful to indicate the vaccination coverage among the study sample, and clarify that the seroprevalence estimates and risk factors are only among those who were unvaccinated

- Abstract: the statement “…thirds of the population were exposed to SARS CoV- 2. Seroprevalence was lowest among individuals >60 years” repeats results already presented and could be removed to allow additional word count;

- Abstract: the undiagnosed fraction (or proportion seropositive without a history of a known diagnosed infection) could be helpful to include in results of abstract

- Methods: clarify the time-period for measuring adherence (assuming adherence to specific covid protocols are time-variant).

- Title: Suggest including “representative sample” in study title. This is a major strength of the study, and highlighting it from the start could be useful.

- Line 41: no need to include the time-stamp as date is sufficient

- Introduction: suggest adding the vaccine coverage (in addition to the absolute number), line 47

- Table 4: unclear what the two models refer to (please describe in methods as well as footnote of Table 4).

6. PLOS authors have the option to publish the peer review history of their article (what does this mean?). If published, this will include your full peer review and any attached files.

**Do you want your identity to be public for this peer review?** For information about this choice, including consent withdrawal, please see our Privacy Policy.

Reviewer #1: No

Reviewer #2: **Yes: **Sharmistha Mishra

---

## [Decision Letter · Decision Letter 1]

3 Apr 2023

Modelling SARS-CoV-2 antibody seroprevalence and its determinants in Ghana: a nationally representative cross-sectional study.

PGPH-D-22-01388R1

Dear Dr. Owusu Donkor,

We are pleased to inform you that your manuscript 'Modelling SARS-CoV-2 antibody seroprevalence and its determinants in Ghana: a nationally representative cross-sectional study.' has been provisionally accepted for publication in PLOS Global Public Health.

Best regards,

Julia Robinson

Executive Editor

Reviewer Comments (if any, and for reference):

Reviewer's Responses to Questions

**Comments to the Author**

1. If the authors have adequately addressed your comments raised in a previous round of review and you feel that this manuscript is now acceptable for publication, you may indicate that here to bypass the “Comments to the Author” section, enter your conflict of interest statement in the “Confidential to Editor” section, and submit your "Accept" recommendation.

Reviewer #2: All comments have been addressed

2. Does this manuscript meet PLOS Global Public Health’s publication criteria? Is the manuscript technically sound, and do the data support the conclusions? The manuscript must describe methodologically and ethically rigorous research with conclusions that are appropriately drawn based on the data presented.

Reviewer #2: Yes

3. Has the statistical analysis been performed appropriately and rigorously?

Reviewer #2: Yes

4. Have the authors made all data underlying the findings in their manuscript fully available (please refer to the Data Availability Statement at the start of the manuscript PDF file)?

Reviewer #2: Yes

5. Is the manuscript presented in an intelligible fashion and written in standard English?

Reviewer #2: Yes

6. Review Comments to the Author

Reviewer #2: Thank you for the revision and thoughtful revisions in light of the suggestions.

7. PLOS authors have the option to publish the peer review history of their article (what does this mean?). If published, this will include your full peer review and any attached files.

**Do you want your identity to be public for this peer review?** For information about this choice, including consent withdrawal, please see our Privacy Policy.

Reviewer #2: **Yes: **Sharmistha Mishra
